



# Wear test programs for roller-type pitch bearings of wind turbines

Matthias Stammler[1]

[1]Large Bearing Laboratory, Fraunhofer IWES, Hamburg, 21029, Germany

**Correspondence:** Matthias Stammler (matthias.stammler@iwes.fraunhofer.de)

**Abstract.** Pitch bearings are critical for the safe and efficient operation of wind turbines. They connect the rotor blades to the rotor hub and allow for pitching movements that control loads and rotor speeds. While four-point contact ball bearings have been dominant in the past, three-row roller bearings are increasingly used in current designs due to their higher load capacity at the same diameter. Wear of the raceways is one of the possible damage mechanisms in pitch bearings. As roller bearings differ significantly from previous designs and because the operational conditions of wind turbines differ from other industrial applications, it is a reasonable de-risking exercise to undergo wear tests prior to the commissioning of such bearings. This study outlines a process for developing a wear test program based on aero-elastic simulation data and wind speed measurements. The process is then applied to an example roller bearing. The final program covers both standstill conditions and pitch cycles. The first is the main addition to former approaches. With existing test rigs and a reasonable budget and timeline, the program can be executed.

## 1 Introduction

The objective of this study is to develop a wear test program for roller-type pitch bearings. Pitch bearings of wind turbines are large slewing bearings that connect each rotor blade with the rotor hub (Burton, 2011). To minimize deformations of the bearing rings, additional stiffening elements such as plates or rings are often employed. There are various commercially available design types for these bearings, as listed in (Stammler, 2020). However, to the best of the author's knowledge, there have only been two publications on wear testing results for pitch bearings with a diameter greater than 2 m: Behnke and Schleich (2022) reported on tests with several four-point contact ball bearings with a pitch diameter of 2.3 m, while Stammler (2020) presented results for two four-point contact ball bearings with a pitch diameter of 4.69 m. Both publications report on short-term tests with static axial loads, and both included results for constant amplitudes and protection runs. Protection runs refer to longer pitch movements embedded within sequences of shorter movements which protect the raceway from wear, as explained in (Stammler et al., 2019a; Stammler, 2020).

Stammler (2020) and later Song and Karikari-Boateng (2021) have outlined long-term wear test programs for pitch bearings. Stammler (2020) applies sequence recognition to find wear-critical series of pitch cycles in time series data. These series, along with any protection runs, are then combined based on seasonal profiles to generate time series that represent the full lifetime of the turbine. Meanwhile, Song and Karikari-Boateng (2021) use a similar approach to recognize wear-critical parameters in individual cycles but do not consider the sequence of these cycles or account for possible protection runs. While the approach





outlined in (Stammler, 2020) yields useful results for pitch cycles, it does not take into account standstill conditions. Therefore, further research is necessary to incorporate standstill conditions into wear testing programs for pitch bearings.

Over the past few decades, several wear tests with constant amplitudes, either as angular amplitudes of the raceways or as load amplitudes, have been published. However, only a minority of these publications cover wear in line contacts. de La Presilla et al. (2023) did a review of oscillating bearing with both point and line contacts.

One early study by Pittroff (1961) subjected cylindrical roller bearings to radial load oscillations. An off-axis weight rotated on a shaft and varied the individual contact loads between $0\,\mathrm{MPa}$ and $<500\,\mathrm{MPa}$. Among other parameters, Pittroff reported on the ability of pretension to reduce the wear damages of the raceways.

In another study, Breward (1973) showed false brinelling damages of oil-lubricated roller bearings. He suggested empirical calculations for the rolling contact fatigue lifetime that take account of the depth of false brinelling damages.

Njoya (Njoya, 1982) used different cage configurations for needle bearings of gearboxes to prevent wear on the rings by forcing a cage movement for operating points where the inner and outer ring have the same speed.

Föhl and Sommer (FVA, 1988, 1991) conducted wear tests on oil-lubricated axial roller bearings (type 81212) to understand

the influence of tribolayers. The test load was $80\,\mathrm{kN}$ and the calculated contact pressure was $1890\,\mathrm{MPa}$. The test rig, a FE8 as defined in (DIN, 1999), rotated the bearing with $7.5\,\mathrm{rpm}$. Föhl and Sommer showed the influence of different additives on wear creation and the beneficial influence of sulfur and phosphate.

Wolf et al. (FVA, 2007) conducted tests on type 81212 bearings using a FE8 test rig and commercial oil for gearboxes and combustion engines. The bearings rotated with $7.5\,\mathrm{rpm}$. Wolf et al. identified a lower load region that does not cause wear and

a higher load region that results in wear.

Schadow (FVA, 2010) tested type 32005 tapered roller bearings to identify the influence of different parameters on false brinelling damages. He used various greases. and the test rig allowed testing at different levels of temperature. The application of dynamic axial, radial, and combined loads caused only slight damages on the bearing raceways. The maximum load amplitude was 25% of the initial static load which was at one third and one tenth of the load capacity of the bearings. Small

oscillating angles under static and dynamic loads caused similar damages to the raceways. As all damages appeared similar, it was not possible to determine the influence of individual test parameters on the damages. The oscillation amplitudes were very small in comparison to wind turbine applications.

In recent studies, Lin et al. (2022) found that zinc dialkyldithiophosphate (ZDDP) additives in grease can effectively prevent wear damage on raceways of cylindrical roller bearings subjected to radial loads and superimposed vibrations.

Stammler et al. described the oscillating behavior of pitch bearings in (Stammler and Poll, 2014), (Stammler et al., 2018, 2019b) and evaluated the influence of different controller types on the oscillating cycles of pitch bearings. They distinguished between Collective Pitch Control (CPC) and Individual Pitch Control (IPC), with the former controlling all three blades simultaneously to adjust the turbine's power output, and the latter controlling the blades individually to limit bending moment variations and reduce fatigue loads on the turbine structure. IPC increases the pitch activity significantly (Stammler et al., 2019b; Requate

et al., 2020). This reduces the bearing's rolling contact fatigue life expectancy. As the loads are highest close to rated wind speed (Burton, 2011), limiting IPC activity at speeds much below rated wind speed is a reasonable approach to reduce wear





and prolong bearing life (Bottasso et al., 2014). This approach may lead to the pitch bearing remaining in rotatory standstill under most operating conditions below rated speed.

Most full-scale test rigs for pitch bearings are operated by commercial businesses. Publications on the rigs are limited to
advertising videos, photos in company reports, and press releases. Examples include (Shanghai Oujikete, 2012; Vries, 2013; Lüneburg et al., 2014; Rollix, 2017; Fischer and Mönnig, 2019; Fangyuan, 2019; Vries, 2019). To the knowledge of the author, there are currently no publicly available test programs or results, even though several test rigs are in operation and roller-type pitch bearings play an increasingly important commercial role. Therefore, developing a test program for full-scale roller-type pitch bearings must address several uncertainties, which will be highlighted in the following sections.

The remainder of this article begins with Section 2, which discusses definitions and damage mechanisms. The following Section 3 describes the process that leads from the data input to the test program output. The data input includes aero-elastic simulation time series, wind speed measurements, and bearing data, while the output consists of test time series in six degrees of freedom. Section 4, which covers application and results, uses the data of the IWT (IWES Wind Turbine) 7.5-164 reference turbine to create a test program for the BEAT6.1 (Bearing Endurance and Acceptance Test) rig of Fraunhofer IWES.

## 2   Definitions

Three-row roller pitch bearings have two axial and one radial row. Figure 1 shows a section view of such a bearing. As the axial loads dominate in pitch bearing operation, the axial rollers are significantly larger than the radial ones. In operation, only one of the axial rollers at the same circumferential position bears the load. One of the rings is typically a split, c-shaped ring, as seen in Figure 1, where the outer ring is split. The bearings are manufactured with little or no preload of the rollers. The
tensioning of the bolts of the c-shaped ring introduces the preload of the rollers. Therefore, the pretension of the bolts is crucial for the reliability of this bearing type.

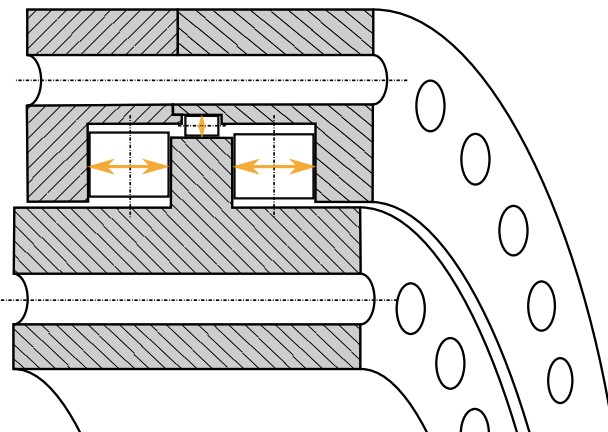

**Figure 1.** Three-row roller pitch bearing (Stammler, 2020)



Any loads and pitch movements in this study use the coordinate system shown in Figure 2 with the exception of the pitch angle $\theta$, which is positive in mathematically negative sense. This convention is industry standard, and most turbine simulation time series follow it.

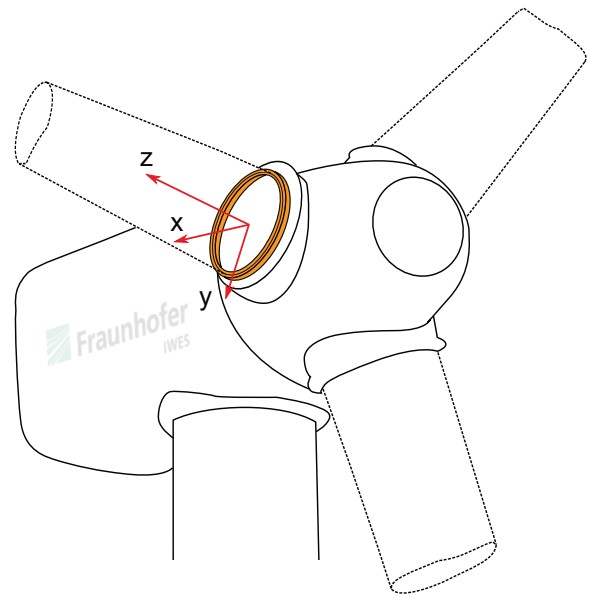

**Figure 2.** Pitch bearing coordinate system (fixed to rotor hub) (Stammler, 2020)

The test program aims at evaluating wear of the raceways. Pitch bearings can also fail for other causes. The author gives an overview of possible damages of pitch bearings in (Stammler, 2020). Within the context of this study, surface wear damages of raceways are distinguished into Standstill Marks and False Brinelling. Figure 3 shows the characteristics of these damages. $x$ denotes the distance covered by the contact center on the raceway during an oscillation, $2b$ the width of the contact area. $Q$ is the load of the contact element.

Standstill marks and false brinelling are both possible in pitch bearing operation. Standstill marks can occur when the bearing rings are in rotary standstill and subject to oscillating loads with $Q_{\min} > 0$ or when they undergo oscillating movements with small amplitudes of $x/2b < 1$. Load oscillations can be caused by rotor rotation and wind fluctuations. Turbine controllers can also cause oscillating movements with $x/2b < 1$, but they have limited positive effects on load and power. To prevent these cycles, turbine controllers can be adjusted. Torsional moments at the blade root that are higher than the bearing friction

torque can also cause these movements, resulting in the blade's center of gravity being off-axis and oscillating around zero. The appearance of standstill marks has a characteristic undamaged central area. False brinelling, on the other hand, can be caused either by oscillating movements with an amplitude ratio $x/2b > 1$, a load oscillation with $Q_{\min} = 0$ or a combination of both. False brinelling does not show an undamaged central area. At more progressed states and with abrasive mechanisms at work, false brinelling damages appear visually like the indents of Brinell tests. Another possible reason for wear of the raceways are



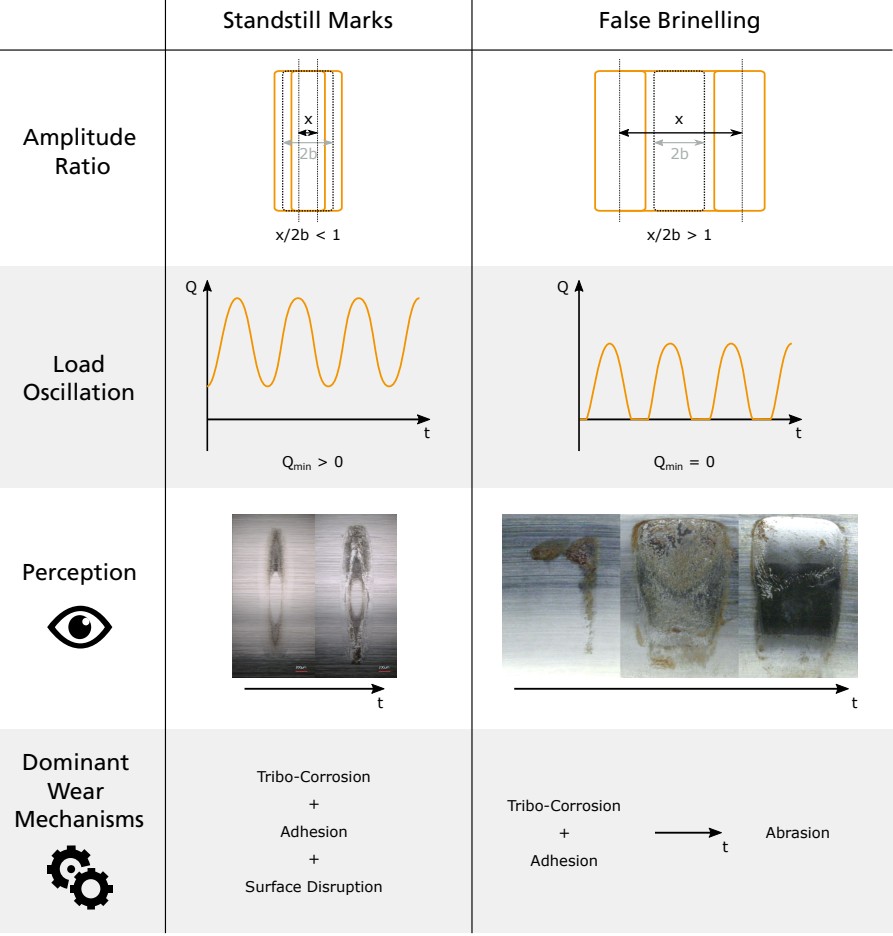

**Figure 3.** Raceway wear damage modes of oscillating bearings

(©wear marks left Markus Grebe, ©wear marks right Sebastian Wandel, both with permission)

movements of the rollers perpendicular to their rolling direction. Loads can cause deformations of the bearing rings, which affect the load distribution along the major axis of the contact and cause these movements. These damages will appear similar to false brinelling but the dominant mechanism is fretting.

The author is not aware of any calculation models that can predict wear on raceways. Instead, Wandel et al. (2022) propose a starvation number that takes into account various parameters, such as oscillation, contact, and lubrication, to assess the risk

of wear.

When wear conditions are met, surface alterations can appear after less than 1000 cycles, and brownish-red corrosion products are clearly visible after at least 5000 cycles (Schwack et al., 2017; Wandel et al., 2023). Effective protection runs are one means to prevent wear on raceways. They can be any longer rotatory movement, whether steady or interrupted. While the existence of protection runs is supported in by experimental evidence (Stammler et al., 2019a; Behnke and Schleich, 2022),

the parameters that determine an effective protection run are not yet fully understood. An effective protection run should be a longer movement, but the exact required amplitude ($\theta_p$) is unknown, and it should occur after several smaller pitch cycles, although their maximum number is also unknown. Results from Stammler et al. (2019a) show effective protection runs for up to 100 cycles in between them. Furthermore, the effectiveness of protection runs depends on factors such as the lubricant used (Schwack et al., 2020, 2021) and cage design (Wandel et al., 2022). These factors will be taken into account in the later test program, and uncertainties will be addressed using a sensitivity analysis.

## 3 Methods

### 3.1 Input data

Stammler (2020) describes the design process of test parts that emulate the interfaces of the pitch bearing. They act as replacement of rotor blade and rotor hub in a more compact test environment and allow dynamic tests with reasonable deformations and energy costs. For this study, these parts and the BEAT6.1 test environment are taken as given. The test profile creation then needs the following inputs:

- Bearing data

  This includes a bearing drawing, the rolling body properties as diameter $d_{\mathrm{rb}}$ and the length $l_{\mathrm{rb}}$, and number of rolling bodies.

- Load simulation output

  This contains time series of all fatigue load cases defined in IEC61400-1 (IEC, 2019). The signals must contain pitch angle $\theta_{\mathrm{sim}}$, blade root bending moments $M_{\mathrm{x}}$ and $M_{\mathrm{y}}$, forces $F_x$, $F_y$, and $F_z$, and information about the mean or current wind speed.

- Wind speed measurement data

  The wind speed measurement data must contain values of at least one year, measured at or close to hub height. Mean values for 10miñ intervals are sufficient.

### 3.2 Comparison of simulation and measurement

The accuracy of test results depends on the quality of input data. To ensure the reliability of simulation data, it is recommended to compare the simulation results with measurement data collected under similar wind conditions. However, in cases where measurement data for the bearing under test is not available prior to testing, comparison with data from similar existing turbines can be used as an alternative approach. This in lieu comparison can be considered reasonable if the simulation methods are the same.



### 3.3 Calculation of additional time series signals

Pitch bearings are typically driven by either hydraulic cylinders or combinations of electrical motors and gearboxes. In the
case of the latter, the gearbox has a geared interface to the bearing with a circumferential backlash denoted as $j_t$. During the
assembly of the bearing, it is not possible to directly measure the circumferential backlash. Instead, the normal backlash $j_n$ is
measured using a feeler gauge. The value of $j_t$ can be calculated as $j_t = j_n/\alpha_n$, where $\alpha_n$ is the normal pressure angle of the
gear pair. If the pitch drive is in a standstill position and a torsional torque $M_z$ is applied in the opposite direction to the last
pitch movement, the pitch bearing can move within the limits of the circumferential backlash $j_t$. In order for the bearing to
move, $M_z$ must exceed the starting torque $M_{t,min}$ of the bearing.

It is important to note that aero-elastic simulations may not necessarily consider the backlash of the pitch gear interface.
The superimposed movement of the pitch bearing at standstill, denoted as $\dot{\theta}_j$, is estimated using Equation 1. The rate of this
movement is assumed to be a fixed value $c_\theta$, whereas in reality, it depends on factors such as the inertia of the deformed blade,
the actual value of $M_z$, and the frictional torque of the pitch bearing. It should also be noted that $\dot{\theta}_j$ is positive in mathematically
negative direction, as per the provided definitions.

$$\dot{\theta}_j = \begin{cases} c_\theta & \text{if} \quad \dot{\theta}_{sim} = 0 \quad \text{and} \quad M_z \leq -M_{t,min} \text{ and} \quad \theta_{flag} = false, \\ -c_\theta & \text{if} \quad \dot{\theta}_{sim} = 0 \quad \text{and} \quad M_z \geq M_{t,min} \text{ and} \quad \theta_{flag} = true, \\ 0 & \text{for all other cases} \end{cases} \tag{1}$$

$\theta_{flag}$ indicates the direction of the last pitch movement. This movement determines which tooth flank is in contact. Equation
2 is executed per each time step $\tau$ if the conditions for any $c_\theta$ are not met.

$$\theta_{flag} = \begin{cases} true & \text{if} \quad \dot{\theta}_{sim,\tau} > 0 \quad \text{or} \quad (M_{z,\tau+1} \geq M_{t,min} \text{ and current } \theta_{flag} = false) \\ false & \text{if} \quad \dot{\theta}_{sim,\tau} < 0 \quad \text{or} \quad (M_{z,\tau+1} \leq -M_{t,min} \text{ and current } \theta_{flag} = true) \end{cases} \tag{2}$$

$\theta_j$ is limited to movements within the backlash $j_t$. In the final test, however, the pitch drive has to overcome the backlash
and move the bearing by $j_t$, so its movement has to be $2 \cdot j_t$. $\theta_j$ is then calculated step-wise as the sum of all previous $\dot{\theta}_j$ unless
its absolute value exceeds $2 \cdot j_t$, in which case it is set to $\pm 2 \cdot j_t$. $\theta_j$ and $\theta_{sim}$ form the final pitch angle $\theta$:

$$\theta = \theta_j + \theta_{sim} \tag{3}$$

The rolling body load $Q$ and the amplitude ratio $x/2b$ have significant influence on wear damages. In the axial rows of
three-row roller pitch bearings, $x$ equals half the distance covered by the rotating ring. Equation 4 calculates the distance $\Delta x$
with $\Delta \theta$ as the pitch angle change between two time steps in $°$, which is common for load simulation outputs. $D_{pitch}$ is the
pitch diameter of the bearing.



$$\Delta x = 0.5 \frac{\Delta\theta \cdot \pi \cdot D_{\text{pitch}}}{360°} \tag{4}$$

The calculation of $Q$ and $2b$ of each individual rolling element is less trivial and commonly relies upon finite-element

analysis. However, the focus of this study is on identifying wear-critical operational conditions, which is less dependent on the exact load $Q$ and more on the combination of operating conditions. Since $Q$ is only used to select operating conditions which are later reproduced exactly by the test setup, empirical formulas can be used to approximate the load at this stage. For the calculation of the $Q_{\text{max}}$, the pitch bearing design guideline (DG03) of NREL (Harris et al., 2009) suggests Equation 5. The radial loads used in (Harris et al., 2009) are omitted here as the axial rollers cannot transfer radial loads. The contact angle $\alpha$ of

the axial rows is assumed to be $90°$, and the bending moment $M$ is calculated as the quadratic sum of the individual bending moments. $Z$ represents the number of rolling bodies per row.

$$Q_{\text{max}} = \left( F_z + \frac{4 \cdot \sqrt{M_x^2 + M_y^2}}{D_{\text{pitch}}} \right) \cdot \frac{1}{Z} \tag{5}$$

The axial rows of pitch bearings are commonly pre-loaded by the bolts of the c-shaped ring. This initial load $q_{\text{ini}}$ needs to be respected in the $Q$ calculations. Under dynamic loads, different rolling bodies are subject to $Q_{\text{max}}$. For a test program creation,

it is more desirable to know the load variations at fixed circumferential positions. A position on the positive x-axis with $y = 0$ is denoted as $0°$. $M_{\text{x}}$ does not influence the load at this position, see Equation 6:

$$Q_{0°} = \left( F_z + \frac{4 \cdot M_y}{D_{\text{pitch}}} \right) \cdot \frac{1}{Z} + q_{\text{ini}} \tag{6}$$

Similarly, $Q$ at $90°$ and $45°$ positions can be calculated, see Equation 7 and Equation 8:

$$Q_{90°} = \left( F_z + \frac{4 \cdot M_x}{D_{\text{pitch}}} \right) \cdot \frac{1}{Z} + q_{\text{ini}} \tag{7}$$


$$Q_{45°} = \left( F_z + \frac{4 \cdot (M_x + M_y) \cdot \frac{1}{\sqrt{2}}}{D_{\text{pitch}}} \right) \cdot \frac{1}{Z} + q_{\text{ini}} \tag{8}$$

The results of Equation 6 to 8 can be negative numbers. It is physically impossible for a rolling body to have a negative load, hence any values below zero must be set to zero, see Equation 9:



$$Q = \begin{cases} Q & \text{if } Q \geq 0, \\ 0 & \text{if } Q < 0 \end{cases} \tag{9}$$

With $Q$ signals added to the input data, it is possible to calculate the contact width $2b$ at the same positions, again for the

axial rows of the bearing. This is done according to Harris and Kotzalas (2007), see Equation 10. $l_{\mathrm{rb}}$ is the length of the rolling

body and $d_{\mathrm{rb}}$ its diameter. $\nu$ is the Poisson's ratio, and $E$ the Young modulus. In this case it is assumed both rolling body and

raceway have the same $\nu$ and $E$.

$$2b = \sqrt{\frac{8 \cdot Q \cdot d_{\mathrm{rb}}}{\pi \cdot l_{\mathrm{rb}}} \cdot \left( 2 \cdot \frac{1 - \nu^2}{E} \right)} \tag{10}$$

Figure 4 shows exemplary output of $Q$ at different positions for turbine operation below rated wind speed for one of the

axial rows. The values are normalized by the maximum value in the simulation file. At $0°$, $Q$ is mainly influenced by the thrust

moment $M_{\mathrm{y}}$ on the blade, resulting in a high mean load level with minor fluctuations. At $45°$ both $M_{\mathrm{y}}$ and $M_{\mathrm{x}}$ influence the

rolling body loads. The variations in load are driven by $M_{\mathrm{x}}$. At $90°$, $M_{\mathrm{x}}$ dominates and since it has negative values, the rolling

body loads become zero with each oscillation. The frequency of $M_{\mathrm{x}}$ matches the rotational frequency of the rotor. It is worth

noting that the DG03 calculation with the quadratic sum of the bending moment shows higher frequencies, reflecting the usage

of absolute values of $M_{\mathrm{x}}$.

Figure 5 depicts the contacts widths $2b$ for the same period, again as exemplary normalized plots. Equation 10 defines a

concave relation between $Q$ and $2b$, which is reflected in Figure 5. The $90°$ position shows the highest variations in $2b$, $0°$

shows almost constant values.

With $2b$ and again following Harris and Kotzalas (2007), the contact pressure $P$ is calculated using as per Equation 11:

$$P = \frac{4 \cdot Q}{2b \cdot l_{\mathrm{rb}} \cdot \pi} \tag{11}$$

This contact pressure allows to estimate the potential of the later test program to impose damages on the raceways.

### 3.4   Standstill program

The underlying idea of a standstill test program is to use the combination of wind speed measurements and simulation data

to find critical operating conditions. Note this part of the test program does not necessarily result in standstill marks, its name

refers to the rotationary standstill of the bearing rings.

The program generates a test plan to replicate the identified operating conditions. The most severe operating condition is

the longest continuous standstill with the same pitch angle under highest $2b$ variations. Wear damages develop over oscillation

cycles (Behnke and Schleich, 2022; Schwack, 2020; Schwack et al., 2021) and are prevented by protection runs (Stammler




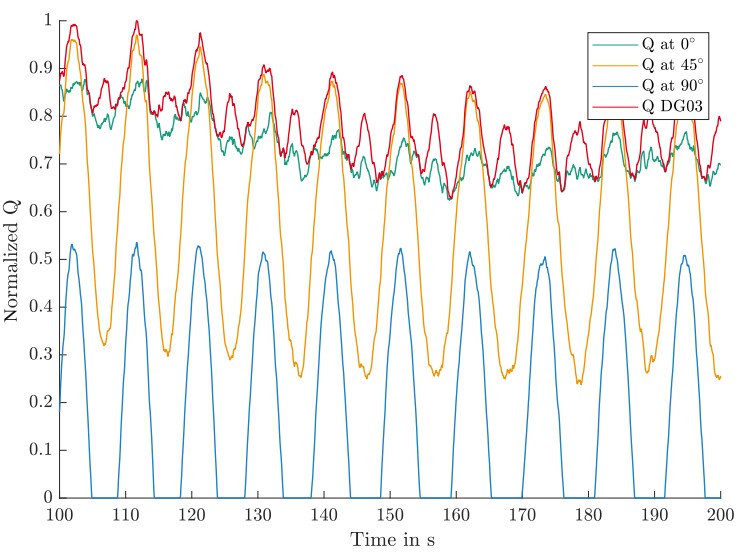

**Figure 4.** Normalized rolling body loads at different circumferential positions for operation below rated speed

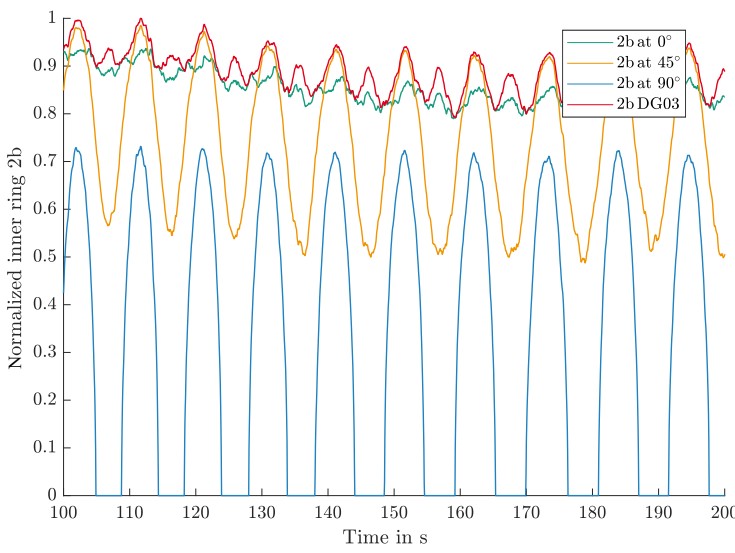

**Figure 5.** Normalized contact widths at different circumferential positions for operation below rated speed





et al., 2019a). Only periods of continuous standstill that meet a minimum cycle number $n_{\min}$ are considered for the test plan. The 10-minute simulation data files hold information on $2b$ variations and short-term standstill conditions. However, for large turbines, it may not be possible to reach the minimum cycle number within 10 minutes. In such cases, files with effective protection runs can be excluded from further analysis. Very short standstill periods are also excluded as they should at minimum have a few load cycles to be relevant for wear progression. This short-term minimum is denoted $n_{\min,s}$. Figure 5 shows that $M_x$, which has the same frequency $\nu_{rotor}$ as the rotor, dominates variations of $2b$. Equation 12 gives the minimum

short-term standstill time $T_{\min,standstill}$:

$$T_{\min,standstill} = \frac{n_{\min,s}}{\nu_{rotor}} \tag{12}$$

$n_{\min}$ is the long-term minimum number of wear-critical cycles. The cycles stop counting when a protection runs occurs. $n_{\min,s}$ is the short-term minimum of cycles that can contribute to wear if they aggregate with more cycles. It is interrupted by pitch movements below protection run threshold $\theta_p$. The simulation data is filtered for protection runs and standstill times

and finally split into wind speed bins. This renders the highest wind speed which still has standstill times. This wind speed is called 'focus speed' in the following. If protection runs occur below focus speed, the files containing them are excluded from the later test run to follow a worst-case philosophy. Pitch bearings spend less time overall (see Figure 6) and shorter periods in standstill with increasing wind speed.

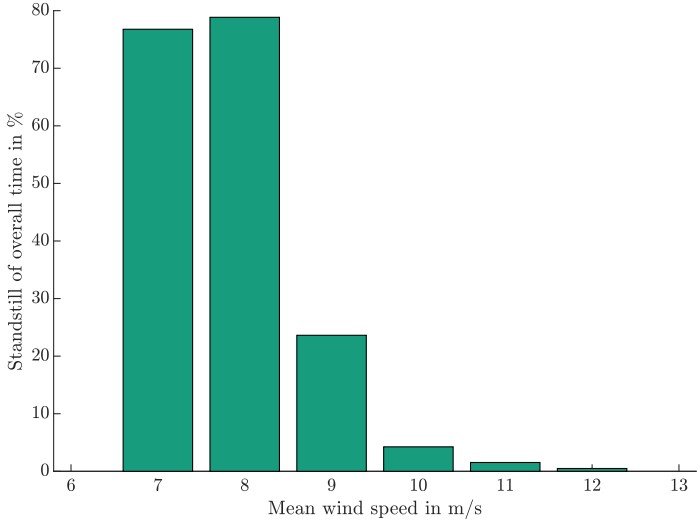

**Figure 6.** Fraction of standstill time over wind speed bin for an example wind turbine





Wind speed measurements allow to put simulation files in mid- and long-term sequences. For a standstill test program, all
speeds at or below focus speed are relevant. Wind speeds bin below cut-in result in idling of the turbine and are taken as end
point for any sequence. The transition from power production to idling causes a long pitch movement which can be assumed
as a protection run. The load cycles can resume during idling due to the gravitational forces of the blade, but the frequency
is magnitudes lower than at power production, resulting in a low number of overall load cycles. Long-term idling of several
months can occur after assembly of the turbine and before it starts power production. Such load cases can be taken into account
in a standstill program but require an acceleration to achieve realistic time frames. This study does not take idling times into
account.

A sequence recognition algorithm as reported in (Stammler, 2020) gets all continuous sequences of 10-min intervals of the
desired speeds and the minimum length defined by $n_{\mathrm{min}}$. It discards any shorter sequences contained in longer ones. Table 1,
taken from (Stammler, 2020), shows the process of this algorithm. The rows are a set of sequence lengths. The cells in green
are the ones the algorithm keeps in the respective step. For a more detailed explanation, see (Stammler, 2020).

**Table 1.** Sequence elimination process

|  | Process step | | |
| --- | --- | --- | --- |
| **0** | **1** | **2** | **3** |
| **Sequence length** | **Sequence length** | **Sequence length** | **Sequence length** |
| 3 | 3 | 3 | 3 |
| 2 |  |  |  |
| 1 |  |  |  |
| 4 | 4 | 4 | 4 |
| 3 | 3 |  |  |
| 6 | 6 | 6 | 6 |
| 5 | 5 | 5 |  |
| 4 | 4 | 4 |  |
| 3 | 3 | 3 |  |
| 2 | 2 | 2 |  |
| 1 | 1 | 1 |  |

For each element of the sequence, a simulation file with the corresponding mean wind speeds is randomly selected. The full
simulation file is used, including any non-standstill times, to reflect realistic turbine conditions.

Between standstill sequences, random files with protection runs are added to the test program. These are full simulation
files as well. At the beginning of every 24 h period of the test, a turbine start is added. Similarly, a turbine stop ends every
24 h period. Transitions prevent signal steps between elements of the test program. They are taken from the simulation data as
described in (Stammler, 2020). Equation 13 renders the overall duration of the standstill part of the test program $T_{\mathrm{TS}}$.





$$T_{\text{TS}} = \sum_{i=1}^{n} T_{\text{seq,i}} + (n-1) \cdot T_{\text{p}} + \sum_{j=1}^{m} T_{\text{start,stop,j}} + \sum_{k=1}^{4n-3+m} T_{\text{trans,k}} \qquad (13)$$

$T_{\text{seq,i}}$ is the duration of sequence $i$ and $T_{\text{p}}$ is the duration of a file with a protection run. $T_{\text{trans}}$ denotes the transition times, and $T_{\text{start,stop}}$ the duration of files with turbine starts and stop. As determination of transition times before composition of the

test program is laborious and provides only limited benefits, they are unknown at the time of test composition. The algorithm thus iterates until all elements form test time series equal or shorter than 24 h.

### 3.5 Pitch cycle program

In (Stammler, 2020), the author suggested a test method that involves taking all wear-critical sequences and protection runs and arranging them in a random or seasonal order to create a comprehensive test program. Both randomized and seasonal test

programs have been developed for the IWT7.5-164 reference turbine, which was equipped with a controller setup from a wind turbine manufacturer. The time series data for the blade roots of this turbine is available at (Popko, 2019).

In order to generate a test program using this method, wear-critical sequences must first be identified in the simulation data. These sequences typically involve oscillations of $\theta$ with similar mean values and amplitudes below $\theta_{\text{p}}$. In the controller setup utilized in (Stammler, 2020), significant activity of the IPC was observed under rated speed conditions, combined with a fixed

lower pitch angle of 0°, resulting in the emergence of multiple wear-critical sequences.

However, if the turbine activates its IPC close to or above rated speed, the CPC is simultaneously activated, causing the mean values of the oscillations to vary significantly. This, in turn, reduces the wear risk associated with ball bearings to a great extent, based on current knowledge.

As there is no knowledge about wear in real-scale roller-type pitch bearings, the test approach for pitch cycles aims at ruling

out this last uncertainty. It identifies the sequences that are most critical, takes the respective simulation files and combines them randomly until 5000 cycles with small amplitudes are reached. The sequences with their sequence lengths $n_i$ are part of simulation files. These files have multipliers $\kappa_i$ which determine their repetitions within the turbine lifetime. These multipliers are changed with a common factor $F$ so $\Sigma_1^5 n_i \cdot \kappa_i \cdot F = 5000$. The final test program is composed of the files with the sequences in random order. The sequence recognition algorithm from (Stammler, 2020) is set to a mean-value tolerance $\theta_{\text{mean,tol}}$ and lower

and upper boundaries of double amplitudes $\Delta\theta_{\text{min}}, \Delta\theta_{\text{max}}$. Of the resultant sequences, the five most critical ones are selected according to the criteria listed in Table 2. Their order represents the importance.

This part of the test does not need a sensitivity analysis, because a variation of parameters would only result in more or less initial sequences to be found. As it is always five sequences picked from the pool of initial sequences, a size variation of the pool does not change the final test program. Similar to the standstill part of the program, transitions as per (Stammler, 2020)

prevent steps at changes of simulation files. Turbine start and stop time series are used at the beginning and end of each 24 h interval.



**Table 2.** Criteria for sequence selection

| Property | Remarks |
|---|---|
| No protection run in file of sequence | Most likely not possible |
| Variance in $\theta_{\mathrm{mean}}$ | Smaller variations are more critical |
| Length of sequence in cycles | Longer sequences are more critical |
| Load level at sequence | Higher loads are more critical |

## 3.6 Sensitivity analysis

In the preceding sections, certain critical parameters of the test program were highlighted, for which no specific values can be ascertained based on the current state of scientific knowledge. Therefore, it is imperative to conduct a sensitivity analysis to
gain an understanding of their influence on the final test program. These parameters are listed in Table 3.

**Table 3.** Parameters of sensitivity analysis

| Property | Unit | Remarks |
|---|---|---|
| $\theta_{\mathrm{p}}$ | ° | Minimum amplitude of effective protection run, determines selection of files for standstill program. Smaller $\theta_{\mathrm{p}}$ result in more files in the standstill program which might make it less wear-critical, but the files used to interrupt standstill sequences might also contain less effective protection runs. |
| $n_{\mathrm{min,s}}$ | - | Short-term minimum number of load cycles at standstill, determines selection of files for standstill program. Smaller $n_{\mathrm{min,s}}$ lead to longer test durations. $n_{\mathrm{min,s}}$ determines $T_{\mathrm{min,standstill}}$. |
| $n_{\mathrm{min}}$ | - | Long-term minimum number of load cycles at standstill. Smaller $n_{\mathrm{min}}$ result in longer test durations. |





$\theta_\mathrm{p}$ and $n_\mathrm{min,s}$ influence the selection of files from the simulation data and define the focus speed. $n_\mathrm{min}$ determines the selection of sequences from the wind speed measurement data.

## 4 Analysis and results

### 4.1 Wind turbine, bearing, and test context

The IWT7.5-164 is a nearshore reference wind turbine. In previous works, the author developed a test program for four-point contact pitch bearings of this turbine (Stammler, 2020). Table 4 lists the main properties of the turbine.

**Table 4.** Main characteristics of the IWT7.5-164, from (Stammler, 2020)

| Property | Symbol | Value | Unit |
|---|---|---|---|
| Hub height | $h_\mathrm{hub}$ | 119.3 | m |
| Specific power (per swept area) | $\psi_\mathrm{rotor}$ | 355 | W/m$^2$ |
| Cut-in wind speed | $V_\mathrm{in}$ | 3 | m/s |
| Rated wind speed | $V_\mathrm{r}$ | 11 | m/s |
| Cut-out wind speed | $V_\mathrm{out}$ | 25 | m/s |
| Minimum rotational speed | $\Omega_\mathrm{min}$ | 5 | rpm |
| Rated rotational speed | $\Omega_\mathrm{r}$ | 10 | rpm |
| Rated tip speed | $V_\mathrm{tip_r}$ | 85.9 | m/s |

In (Stammler, 2020) the pitch controller activates IPC below rated speed, which renders only limited benefits but adds significant wear-critical cycles to the pitch bearing operation. It is reasonable to assume current commercial controllers do not use IPC under rated speed. Requate et al. (2020) compare CPC and IPC version of the IWT7.5-164. For the present work,

the time series of Requate's comparison are split. Below rated speed, the CPC variant is used, at and above rated speed, the IPC variant. The controllers are built for research purposes and are not tuned on the same detail level as commercial controllers. The data consists of power production load cases with mean wind speeds from $3\,\mathrm{m/s}$ to $27\,\mathrm{m/s}$. The simulated time is approximately $165\,700\,\mathrm{h}$ (18.9 years).

The wind speed measurement data is the same as in (Schwack et al., 2021) and stems from a nearshore met mast with a

measurement height of $119\,\mathrm{m}$ which conveniently fits the hub height of the IWT7.5-164. The data spans the time of one year. As the IWT is a reference wind turbine, it has never been built in reality. A comparison of measurement and simulation data is thus not possible for this case study.

Table 5 lists those properties of the three-row roller bearing that are relevant for the test program design.

With these properties, the additional signals of the simulation data are calculated as described above. Table 6 lists exemplary

maximum values, all of which occur at $17\,\mathrm{m/s}$ mean wind speed except for $M_\mathrm{z,max}$ which occurs at $3\,\mathrm{m/s}$. It is important to





**Table 5.** Characteristics of the pitch bearing

| Property | Symbol | Value | Unit |
|---|---|---|---|
| Pitch diameter | $D_{\text{pitch}}$ | 4719 | mm |
| Rolling body diameter | $d_{\text{rb}}$ | 50 | mm |
| Rolling body length | $l_{\text{rb}}$ | 50 | mm |
| Roller count per row | - | 255 | - |
| Normal backlash | $j_{\text{n}}$ | 1 | mm |
| Gear pressure angle | $\alpha_{\text{n}}$ | 20 | ° |
| Starting torque | $M_{\text{t,min}}$ | 30 | kNm |
| Constant pitch speed in backlash | $c_\theta$ | 0.3 | °/s |

note that these maximum values without safety factors. Upon closer examination, it is apparent that $M_{\text{z,max}}$ is very close to the starting torque of the bearing. Moreover, in most operating conditions, $M_{\text{z}}$ takes on negative values.

**Table 6.** Exemplary maximum values

| Symbol | Value | Unit | Wind speed |
|---|---|---|---|
| $M_{\text{z}}$ | 40 | kNm | 3 m/s |
| $M_{\text{xy}}$ | 25 446 | kNm | 17 m/s |
| $Q_0$ | 88.04 | kN | 17 m/s |
| $P$ | 1.608 | GPa | 17 m/s |
| $2b_0$ | 1.394 | mm | 17 m/s |

This characteristic of $M_z$ is due to the turbine design and not necessarily the same for other turbines. Figure 7 shows the movement of the pitch bearing within the backlash. This is at a wind speed of 3 m/s. At higher wind speeds the movement does not occur. Movements within the backlash will thus be very limited in the standstill part of the final test program.

The test run is going to be conducted on the BEAT6.1 rig (Stammler, 2020). This rig is capable of applying dynamic loads in six degrees of freedom while providing realistic interface conditions for the bearing. The target time frame for both standstill and pitch cycle program are 40 days. The aim of the test is to evaluate the risk of wear in standstill and pitch cycle situations. If it is possible within the time frame, the program should also reproduce the standstill situation of the turbine lifetime.

### 4.2 Ranges for sensitivity analysis

The minimum amplitude for an effective protection run is denoted by $\theta_{\text{p}}$. Stammler (2020) reported a protection run of approximately $4°$ or $x/2b = 30$ to be partially effective for a 5 m ball bearing, as the majority of contact tracks on the raceway did not show any damage. This protection run was executed every 30 cycles. In a study by Behnke and Schleich (2022), protection

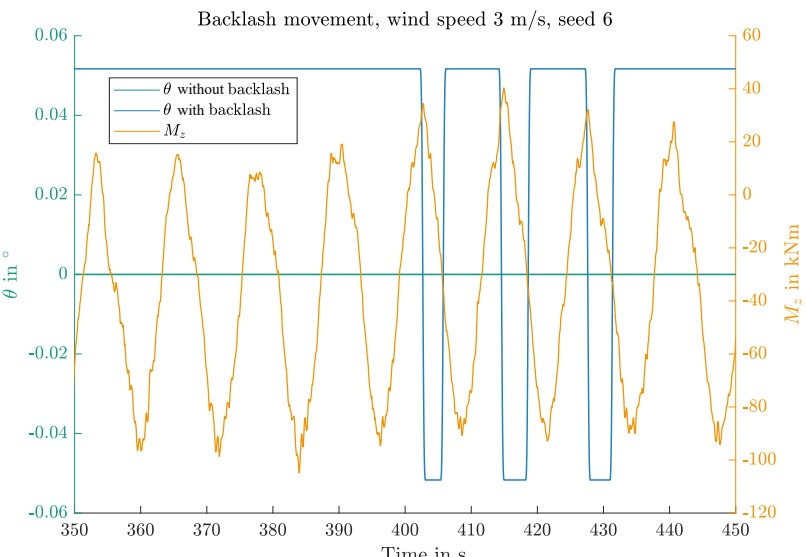

**Figure 7.** Pitch movement in gear backlash due to $M_z$ at $3 \, \mathrm{m/s}$ mean wind speed

runs were performed on a $2.3 \, \mathrm{m}$ ball bearing with a travel of $4°$ or $x/2b = 40$. They reported no wear if the protection run

was executed after 10 or 50 cycles, but observed beginning wear if it was executed after 100 cycles. Based on these limited experiences with other bearing types, the range of $\theta_\mathrm{p}$ for the sensitivity analysis is set between $3°$ and $9°$ for this study.

$n_\mathrm{min,s}$ is assumed at five load cycles at the rated speed of the turbine which is approximately $10 \, \mathrm{rpm}$. $T_\mathrm{min,standstill}$ is then $30 \, \mathrm{s}$. This bases on the assumption that wear cannot be started or progressed with less than 0.5 load cycles, which appears reasonable but is not backed by test data. However, a value for $n_\mathrm{min,s}$ is necessary to avoid identification over very short time

intervals, i.e. $< 1 \, \mathrm{s}$ with no pitch movements as standstill. Larger values for $n_\mathrm{min,s}$ make the subsequent filter stricter and allow less operational scenarios to be incorporated in the final test. To understand the influence of this parameter, it is varied between 0.5 and 10.5 cycles in three steps.

$n_\mathrm{min}$ is set to 250 cycles, which, at rated rotor speed of $10 \, \mathrm{rpm}$, equals 25 minutes. Published data indicate 100 cycles between protection runs to result in no wear on the raceway (Stammler et al., 2019a), and ongoing research shows effective

protection for up to 250 cycles. As the wind speed measurements and simulation data are in $10 \, \mathrm{m\tilde{i}n}$ intervals, the minimum sequences length is set to 30 min or 300 cycles. The range of the sensitivity analysis is set from 200 to 400.

Table 7 lists the parameters and their ranges for the sensitivity analysis. The individual steps result in 27 possible combinations. The sensitivity analysis for $\theta_\mathrm{p}$ and $n_\mathrm{min,s}$ is done with the simulation data, the sensitivity analysis for $n_\mathrm{min}$ with the wind speed data.





**Table 7.** Ranges for sensitivity analysis

| Property | Range | Unit | Steps |
|---|---|---|---|
| $\theta_\mathrm{p}$ | $3-9$ | ° | 3 |
| $n_\mathrm{min}$ | $200-400$ | - | 3 |
| $n_\mathrm{min,s}$ | $0.5-10.5$ | - | 3 |

### 325  4.3  Standstill program results

Figure 8 displays the maximum pitch angle deltas for each simulation file, providing insight into the applicability of the sensitivity range for $\theta_\mathrm{p}$. For wind speeds up to $7\,\mathrm{m/s}$, $\theta$ remains unchanged. At $9\,\mathrm{m/s}$, the CPC controller has to limit the power output of the turbine at some occasions. At $11\,\mathrm{m/s}$ and above every file has maximum pitch angle deltas above the maximum value of the $\theta_\mathrm{p}$ range.

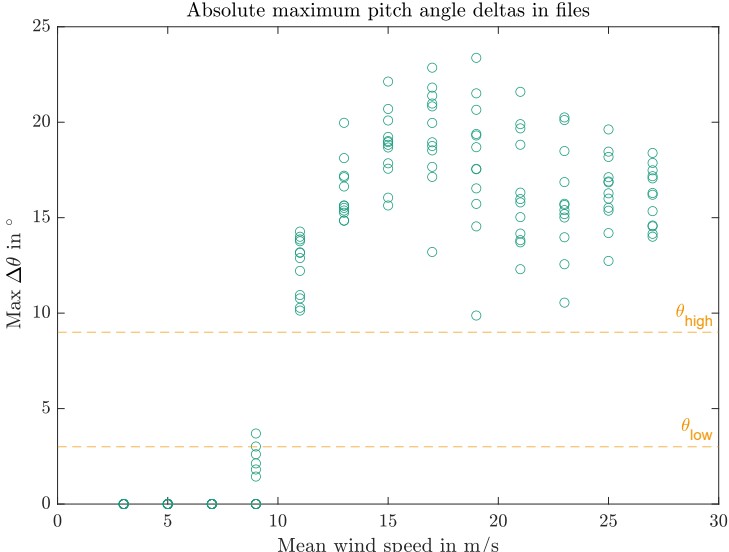

**Figure 8.** Maximum pitch travel $\Delta\theta$ over mean wind speed

Since the different values for $\theta_\mathrm{p}$ will not influence the focus speed, $\theta_p$ is set to 6°. With $\theta_\mathrm{p}$ fixed, $n_{min,s}$ is varied and the simulation data is filtered for standstill times. Figure 9 illustrates the remaining simulated times and simulation files over wind speed. The left side shows $n_\mathrm{min,s}=0.5$, the center $n_\mathrm{min,s}=5.5$, and the right side $n_\mathrm{min,s}=10.5$. This figure allows to select focus speeds. The differences are negligible and the assumed value of $n_\mathrm{min,s}=0.5$ is maintained. The resulting focus wind speed is $9\,\mathrm{m/s}$.

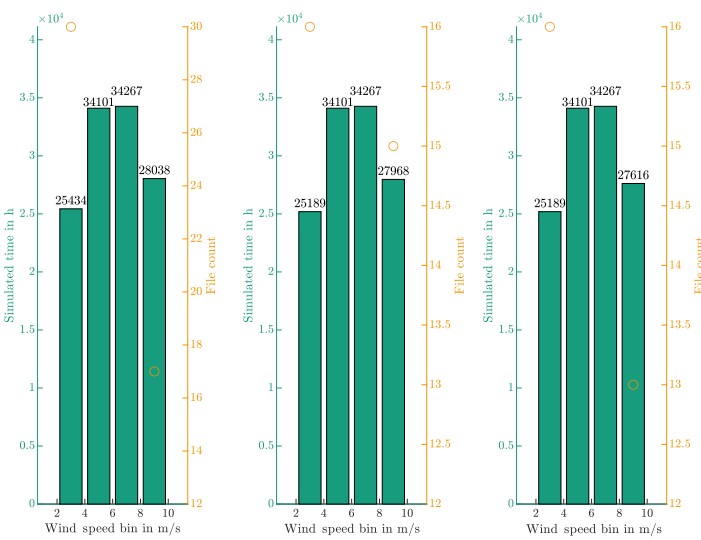

**Figure 9.** Standstill time and files depending on $n_{\mathrm{min,s}}$

The sequence recognition uses the focus wind speed and the three possible values for $n_{\mathrm{min}}$ with the wind measurement data. The expected result are three different overall lengths of $\Sigma_{i=1}^n T_{\mathrm{seq,i}}$. Figure 10 shows an overview of the measurement data with maximum and mean values of 10 min periods. Minimum speeds are not available in this data set, which covers one year. As expected, the wind speeds are below cut-in and above the focus speed at several instances.

Table 8 lists the resulting $\Sigma_{i=1}^n T_{\mathrm{seq,i}}$ of all sequences within the thresholds. $n_{\mathrm{min}}$ has a minor influence on the test duration. As the wind speed data covers one year, the duration would have to be multiplied with 20 to result in an endurance run. Neither 20 nor one year as test input are possible within the target time frame.

**Table 8.** Resulting lengths of sequences for one year in days

| $n_{\mathrm{min}}$ | $\Sigma_{i=1}^n T_{\mathrm{seq,i}}$ in days |
|---|---|
| 200 | 146.4 |
| 300 | 141.8 |
| 400 | 137.6 |

With an endurance run resulting in unrealistically long test times, the target of the test is a risk evaluation. This allows to reduce the input time span, as long as the resultant test has sufficient cycles to result in wear. The wind speed data is split into eight equal parts of which the one with the lowest average speed is selected. Figure 11 shows these 1.5 months of the wind



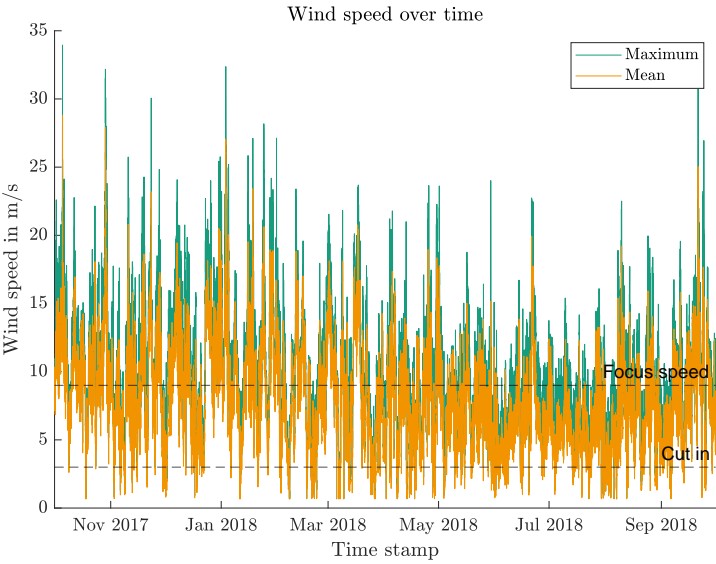

**Figure 10.** Measured wind speed overview

speed and Table 9 lists the overall duration of sequences. All of these fit into the desired time frame. Hence, $n_{\min}$ is set to the lowest value of 200 to integrate as many operational scenarios as possible into the test.

**Table 9.** Resulting lengths of sequences for one year in days

| $n_{\min}$ | $\Sigma_{i=1}^{n} T_{\mathrm{seq,i}}$ in days |
|---|---|
| 200 | 28.8 |
| 300 | 28.1 |
| 400 | 27.6 |

The algorithm described in (Stammler, 2020) generates test time series for the standstill program, resulting in a final duration of 30.7 days. In the test, the maximum values of the simulation data listed in Table 6 remain the same, as the high number of standstill sequences, coupled with a random selection of protection runs, caused all files from the input data set to be part of

the test run. It is worth noting that this is a result of the case study and may not necessarily be applicable to other turbines.

Exemplary pitch angle $\theta$ curves of different test days for a selected duration of 2 hours are shown in Figure 12. The graph clearly displays the difference between standstill times and protection runs.

Figure 13 displays one exemplary transition from a protection run load case to a standstill period. It contains the contact pressure $P$ at different positions and the pitch angle $\theta$. The peak contact pressures at 90°, in blue, are almost constant, whereas



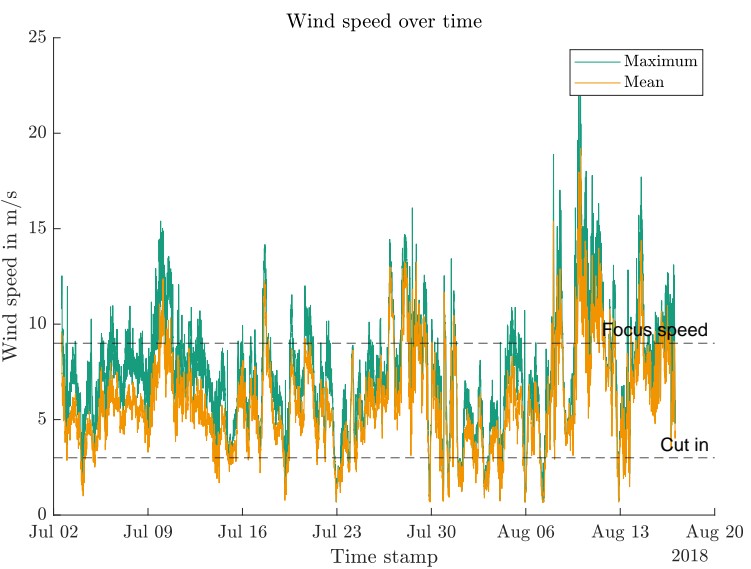

**Figure 11.** Measured wind speed overview, 1.5 months

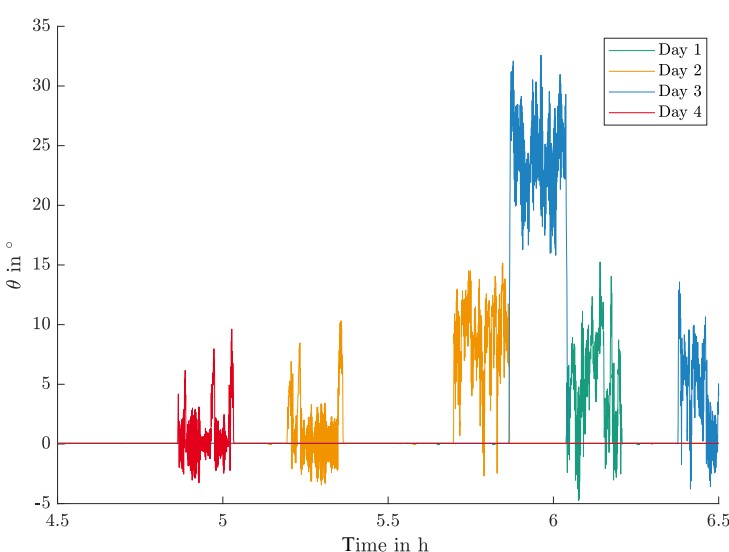

**Figure 12.** Exemplary $\theta$ of the test time series





the mean pressure value at $0°$, in green, drops once the load case of the protection run ends at approximately $392.6\,\text{min}$. The protection run load case stems from a file with higher wind speed and thus higher thrust load $M_y$.

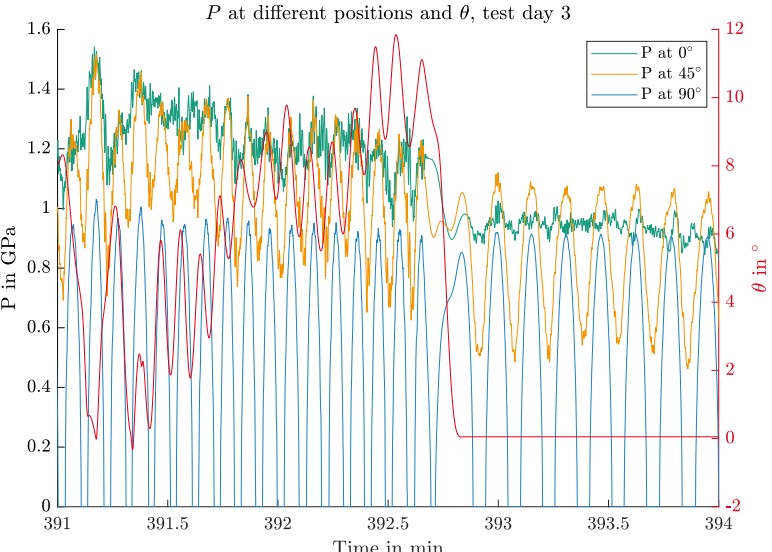

**Figure 13.** Exemplary P and $\theta$ of the test time series

These two exemplary figures show that the final test program combines standstill periods and protection runs. A cycle count of $P$ at $90°$ returns 275219 cycles with an average frequency of $0.18\,\text{Hz}$.

### 4.4  Pitch cycle program results

The algorithm described in (Stammler, 2020) identifies sequences in the pitch movements using the criteria listed in Section 3. Table 10 shows the values for this analysis. To avoid counting dithering moves, $\Delta\theta_{\text{min}}$ is set to 0.05. Similarly, to avoid incorporating partially effective protection runs into the sequences, $\Delta\theta_{max}$ is set to 3. $\theta_{\text{mean,tol}}$ is set to $1°$, which ensures the cycles affect the same area of the raceway. Since the final sequences are selected from all those found using these criteria, there is no need for a sensitivity analysis for this part.

**Table 10.** Conditions for pitch sequence analysis

| Property | Value | Unit |
|---|---|---|
| $\theta_{\text{mean,tol}}$ | 1 | ° |
| $\Delta\theta_{\text{min}}$ | 0.05 | ° |
| $\Delta\theta_{\text{max}}$ | 3 | ° |





The analysis returns two sequences. As the five most critical should have been selected, it is not necessary to apply the criteria from Table 2 and the two sequences are maintained for the test. Figures 14 and 15 show $\theta$ of these sequences in green. The dotted black lines are $\theta$ of the entire file. It can be seen that in both cases assumed protection runs, i.e. pitch movements spanning more than $6°$, are in proximity of the sequence. The orange lines show the blade root bending moment. Both sequences occur at $11\,\mathrm{m/s}$ mean wind speed.

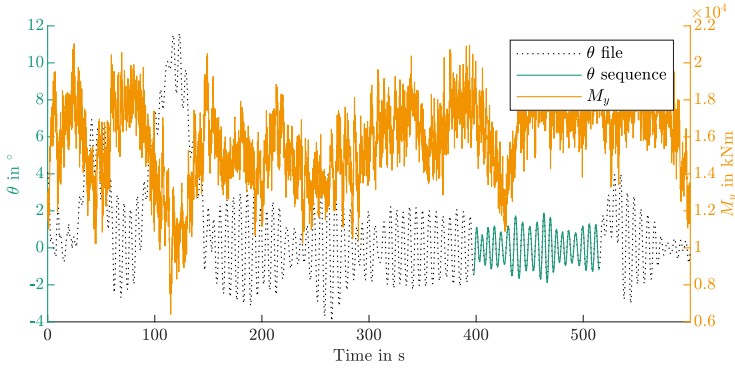

**Figure 14.** Sequence 1 for pitch cycle program, mean wind speed $11\,\mathrm{m/s}$

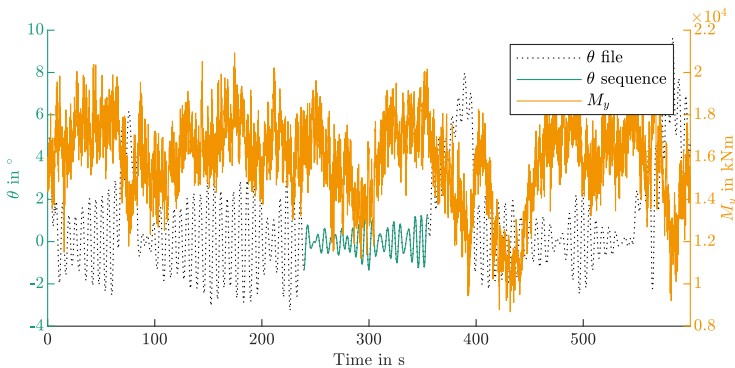

**Figure 15.** Sequence 2 for pitch cycle program, mean wind speed $11\,\mathrm{m/s}$

Table 11 lists the properties of the two sequences. Their few cycles and the proximity to assumed protection runs results in a possibly low wear risk. However, this evaluation is based on experiences with ball bearings, hence it is reasonable to conduct a practical evaluation for roller bearings.





**Table 11.** Properties of sequences for pitch cycle program

| Sequence | Cycles | Mean travel in ° | Mean frequency in Hz |
|----------|--------|------------------|----------------------|
| 1 | 15.5 | 2.19 | 0.33 |
| 2 | 17 | 1.55 | 0.33 |

Similar to the standstill program, the algorithm from (Stammler, 2020) then builds test time series. Each sequence is repeated
154 times which results in 5005 critical cycles in the test program. The final duration of the pitch cycle program are 2.1 days.

The contact pressure $P$ at $0°$ is in the range from $1\,\mathrm{GPa}$ to $1.5\,\mathrm{GPa}$ for the entire test. Figure 16 displays $\theta$ and $M_\mathrm{y}$ of the
second test day at different zoom levels. The top graph shows the maximum pitch travel $\Delta\theta$ each 15 minutes. The red and
green rectangle indicate the time window of the mid resolution plots in the center of the Figure. In each of these, the rectangles
again indicate the time window of the lowermost plots which show one minute each.

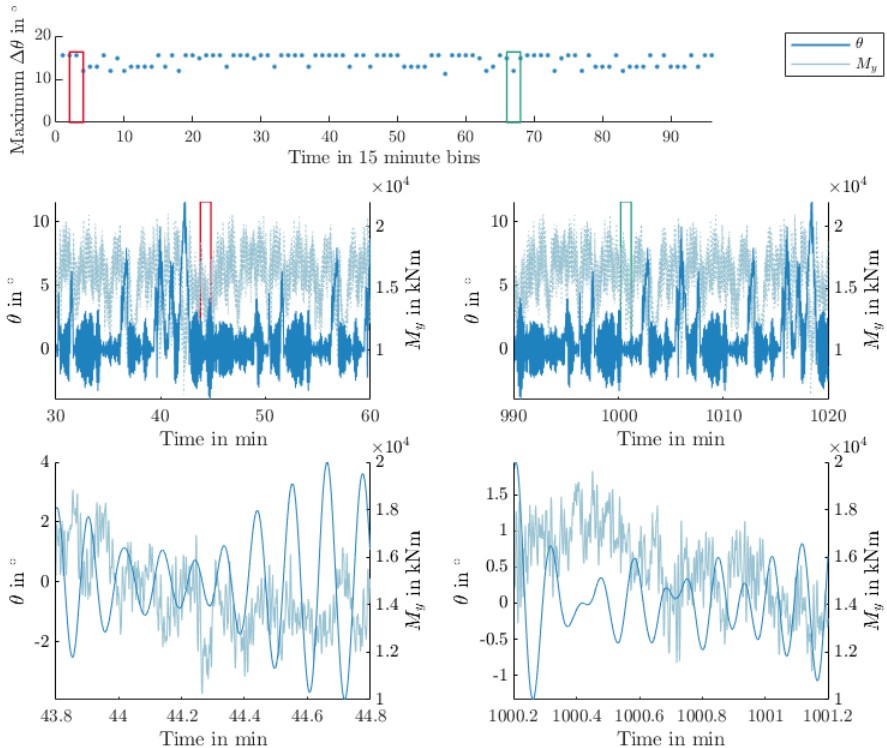

**Figure 16.** $\theta$ and $M_\mathrm{y}$ at different zoom levels

Ratios of $x/2b$ are used in several publications to describe amplitudes of oscillations. In case of time series with dynamic
loads, these ratios have to be calculated per each time step. Figure 17 shows an exemplary results for $30\,\mathrm{min}$ of the pitch cycle





test run. The values are the cumulated sums of step-wise calculations of $\Delta x/2b$ for the $0°$ and the $90°$ position, calculated with equations 4 and 10. The grey dotted line shows cumulated $\Delta x$. $\Delta x$ obtains negative values when $\Delta\theta$ is negative. The increase of the values at $90°$ is to be noted and caused by continuously different load situations on the upward and downward slopes of the pitch movement. This changes the values of $2b$ and with it the values of $\Delta x/2b$. The values at $0°$ increase as well, but at

a significantly lower rate. In tests with steady load situations, such increases do not occur. The values of $x/2b$ for individual cycles during the critical sequences are at approximately 20, which is below the threshold of protection runs mentioned above.

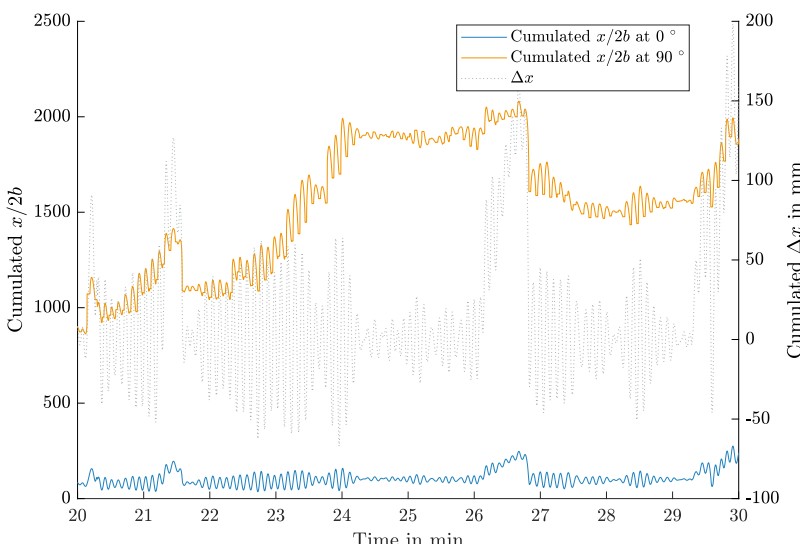

**Figure 17.** Cumulated $x/2b$ and cumulated $\Delta x$

## 5   Discussion, conclusions, and outlook

This study presents a new approach for creating a wear test program specifically designed for roller-type blade bearings. The program covers two distinct operational scenarios: the standstill of the bearing under rated speed and the active pitch cycles

caused by the IPC. This study builds on previous works by including the operation under bearing standstill, which was not covered in previous wear test programs for pitch bearings. To achieve this, a combination of aero-elastic simulation data and wind speed measurement data is used to capture short-term and long-term operational characteristics of the bearing.

    In pitch bearings with geared drives, torsional moments at the blade root can cause movement within the gear backlash. This phenomenon is commonly not accounted for in aero-elastic simulations. To address this issue, this study proposes a method to

include such movements in the time series. It further suggests to double the amplitude of these movements and use the output as setpoints for the pitch drive. This approach can force the bearing into a movement that equals the one caused by blade



root moments $M_\mathrm{z}$ and overcomes the pitch drive's own backlash. By including these movements in the test program, the wear behavior of the pitch bearings under realistic operating conditions can be assessed more accurately.

To take account of the lack of operational experiences with roller-type blade bearings, the process includes sensitivity analysis of three key parameters: the travel of an effective protection run $\theta_\mathrm{p}$, the short-term minimum load cycles that contribute to wear progression $n_\mathrm{min,s}$, and the long-term minimum cycles before interruption by a protection run $n_\mathrm{min}$. The proposed method is demonstrated on the IWT7.5-164 reference turbine, using a mix of simulation results from CPC and IPC variants to mimic current commercial controller strategies. Results show that within the range of 3–9° and 5–10 cycles for $\theta_\mathrm{p}$ and $n_\mathrm{min,s}$, respectively, these parameters have no significant influence on the wear test program. The parameter $n_\mathrm{min}$ has a limited influence on the test duration. However, it should be noted that these results are specific to the chosen turbine and may not necessarily apply to other turbines.

The standstill part of the program follows the initial idea of combining standstill periods and protection runs to interrupt them. It has a duration of approximately 30 days. During this period, the contact pressures are maintained at 0.9–1.5 GPa for most of the test time at 0°, while ranging from 0 GPa to approximately 1 GPa at 90°. Although these pressures exceed those used in Pittroff's experiments (Pittroff, 1961), it is worth noting that Pittroff's experiments were performed with less advanced lubricants, and it can be questioned whether the load levels at the 90° position are sufficient to cause noticeable wear damages. As Lin et al. (2022) have pointed out, ZDDP additives, commonly used in modern commercial lubricants, play a crucial role in preventing wear under standstill conditions. The load cycle count in this study is 275000, which is comparable to that used in Pittroff's and Lin's (Lin et al., 2022) experiments involving standstill and load oscillations.

The pitch cycle part of the program follows the procedure described in (Stammler, 2020) with some modifications. In the present study, the selection of sequences is done manually, and the number of cycles is limited to 5000, making it a risk assessment rather than an endurance test. Based on experience from ball bearings, the probability of wear is expected to be very low, as the program includes over 300 protection runs. Additionally, an analysis of $x/2b$ ratios at different circumferential positions indicates potential for different slip regimes, which could cause the rolling body set to move overall and further reduce the probability of wear.

The two program parts have a combined duration of approximately 32 days. The maximum loads and load dynamics can be realized with the planned test equipment.

Future work for this study includes several items. First, the roller bearings described in this study will undergo the entire test program, consisting of both the standstill and pitch cycle tests, on the BEAT6.1 rig. The bearing will be equipped with inductive sensors to monitor the position of the rolling body set throughout the test. Secondly, turbine starts and stops will be included into the test time series, as they were not part of the input data used in this study. Thirdly, the overall load level is low and there is the possibility of upscaling the loads as both the bearing and test equipment can sustain higher loads. Increasing the load level may yield more informative results. Finally, the process of the test program creation will be applied to other turbines in the future.





*Data availability.*   The time series data of the aero-elastic simulations are available from the author upon request. The time series of the final test program are publicly available at Fraunhofer Fordatis repository.

*Author contributions.*   Matthias Stammler: Everything

*Competing interests.*   Matthias Stammler declares there are no competing interests.

*Disclaimer.*   I disclaim this to be a very nice paper.

*Acknowledgements.*   This work was carried out within the Project HAPT2. The funding by the BMWK, Federal Ministry for Economic Affairs and Climate Action (Federal Republic of Germany), under grant number 03EE2033A, is kindly acknowledged. The wear mark photos in Figure 3 are used with permission by Dr. Markus Grebe and Sebastian Wandel, which is kindly acknowledged as well. The author also likes to thank Arne Bartschat and Eike Blechschmidt for gleeful in-depth conversations about the big and little kinks in the overall process. Niclas Requate's time series are the base of the case study in this paper and the author thanks for the permission to use them.

Real-scale blade bearing tests are teamwork. The author counts himself lucky to have superb colleagues at Fraunhofer IWES' Large Bearing Laboratory and thanks them very much for their support in past and future test runs: Nils Thormälen, Heinrich Drath, Oliver Menck, Karsten Behnke, Matthis Graßmann, and Florian Schleich.



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
