# Peer review of "Wear test programs for roller-type pitch bearings of wind turbines"

_Wind Energy Science, 2023_

## Referee Comment (RC1)

**Review**

**General comments**

In my opinion, the publication represents a very useful contribution to scientific progress in the context of WES. It is of high interest to the entire wind power community.

The aim of the study is to develop a realistic wear test program for the rolling bearings of wind turbine blades. The program combines conditions that prevail when the blade bearings are at a standstill due to the vibration load with conditions during the adjustment movement. The angles and time components are determined from real wind measurements and application-oriented load simulations on the reference turbine IWT7.5-164. The developed simulation program is to be used to run original bearings on the original BEAT6.1 blade bearing test rig.

The author explains in great detail how the individual load sequences are derived and evaluated from the real application and how the complete test program is put together from this. The complex CPC and IPC control algorithms are also taken into account. The test program is interrupted by protection runs, which are intended to avoid damage caused by standstill marks.

The procedure is explained very well. All stress components are well justified.

Accordingly, I consider the scientific relevance, the scientific quality and the quality of presentation to be very good.

**Specific comments**

Line 77: The number of rolling elements that can support the axial load depends on the elasticity, stiffness, and fit dimensions. I do not believe that only one rolling element carries the complete load. However, the load on the rolling elements in the load zone will be very different.

Among the references, some from Mannheim should be mentioned in addition to the numerous publications from Hanover. This research group has also contributed much to the understanding of false brinelling or standstill conditions in recent decades.

**Technical corrections**

None.

---

## Author Response (AR1)

**Reviewer #1**

*Review General comments*

In my opinion, the publication represents a very useful contribution to scientific progress in the context of WES. It is of high interest to the entire wind power community. The aim of the study is to develop a realistic wear test program for the rolling bearings of wind turbine blades. The program combines conditions that prevail when the blade bearings are at a standstill due to the vibration load with conditions during the adjustment movement. The angles and time components are determined from real wind measurements and application-oriented load simulations on the reference turbine IWT7.5-164. The developed simulation program is to be used to run original bearings on the original BEAT6.1 blade bearing test rig. The author explains in great detail how the individual load sequences are derived and evaluated from the real application and how the complete test program is put together from this. The complex CPC and IPC control algorithms are also taken into account. The test program is interrupted by protection runs, which are intended to avoid damage caused by standstill marks. The procedure is explained very well. All stress components are well justified. Accordingly, I consider the scientific relevance, the scientific quality and the quality of presentation to be very good.

*Specific comments*

Line 77: The number of rolling elements that can support the axial load depends on the elasticity, stiffness, and fit dimensions. I do not believe that only one rolling element carries the complete load. However, the load on the rolling elements in the load zone will be very different. Among the references, some from Mannheim should be mentioned in addition to the numerous publications from Hanover. This research group has also contributed much to the understanding of false brinelling or standstill conditions in recent decades.

*Technical corrections*

None

**Answer to Reviewer #1:**

The author thanks the reviewer for its time and effort.

Regarding the comment on line 77: The original sentence reads "In operation, only one of the axial rollers at the same circumferential position bears the load." The author admits this wording lacks elegancy, while he would argue it is still correct because of the additional "at the same circumferential position. The new version reads "In operation, only one of the axial rows at the same circumferential position bears the load."

The author agrees with the reviewer about the relevance of the works of the Mannheim tribology center, which have been published before the FVA reports. Also, Mannheim has published numerous important papers since 2006. To the best knowledge of the author, the Mannheim publications do not cover line contact results, however, and have thus not been a part of the reference list. But since this work makes use of the terms "Standstill marks" and "False Brinelling" as firstly brought forward by Grebe, the author added this remark and points to the respective publications.

**Reviewer #2:**

*General Comments*

In this paper, the author sets out an ambitious goal: to outline a process for developing a wear test program for roller-type pitch bearings based on aero-elastic simulation data and wind speed measurements. In that respect, it provides a valuable potential contribution to the wind industry, as the only strict design requirement currently stated in IEC 61400-1 relates to the static load capacity; whereas, a much more vague requirement that "…consideration shall be given to the potential effect of insufficient lubrication due to small movement…" relates to wear. The subject of this paper could indeed one day lead to greatly improved design and/or design verification requirements in IEC 61400-1 or an equivalent standard.

*Specific Technical Comments*

Line 103: I recommend that the authors have a look at Brinji, O., K. Fallahnezhad, and P.A. Meehan. 2020. "Analytical Model for Predicting False Brinelling in Bearings." Wear 444–445: 203135. doi: https://www.sciencedirect.com/science/article/pii/S0043164819313699. Additionally, there may be more recent works that cite this (see https://www.sciencedirect.com/science/article/pii/S0043164819313699#section-cited-by), which I would recommend having a quick look at as well. I will admit I have only briefly scanned this work and need more time to digest it myself, but it might change how the sentence at line 103 is written. Having said that, these works relate to false brinelling, rather than the more general "wear" described in line 103. I invite the author's thoughts here in either agreement or disagreement.

Line 181: I certainly agree that the applied load can overcome the preload such that the roller is in clearance and no longer carries a load, thus Equation (9) is true. However, I have to wonder if this load Q that is set to 0 at this position, then must be carried by the roller opposite (180 degrees) from this position? Is that accounted for in the formulation? Effectively, we have a system of parallel springs that can only carry load in one direction. I invite the author's thoughts here as well.

*Minor Clarifying Comments*

Line 20: For the more general wind audience, I recommend adding something like the following "…, which protect the raceway from wear by redistributing the grease in the pitch bearing, as explained…" to this sentence.

Line 29: I recommend adding "…decades, results of several wear tests…". I also believe many of the sentences in lines 29-54 can/should be made into a single paragraph.

Lines 78-79: I will admit I don't have experience with this type of bearing, so I don't know the nomenclature. However, I think it's a little clearer to say "One of the rings is typically C-shaped and split into two halves, as shown in Figure 1, where the outer ring is split."

Lines 80: I think it's a bit clearer to say for the general wind audience "When the blade, pitch bearing, and hub are assembled, the tensioning of the bolts of the c-shaped ring introduces the preload of the rollers." Alternatively, it is typical that the outer ring is stationary with the hub, so one could say "When the pitch bearing and hub are assembled, …"

Line 85: Again for a more general audience, it might be worthwhile to add "…evaluating wear of the raceways, which are the surfaces of the bearing rings that contact the rolling elements." It might even be worth it to label the inner ring, outer ring, and raceway surfaces in Figure 1.

Line 96: It feels like the sentence "The appearance of standstill marks has a characteristic undamaged central area" is at first a bit confusing (i.e. a mark that leaves no mark). I think something like "Standstill marks typically occur at the roller ends, often separated by an undamaged central area as shown in Figure 3" is clearer.

Line 131: It appears there are some typos here with the tilde. Same for line 320.

**Answer to Reviewer #2:**

The author thanks the reviewer very much for the helpful comments.

In the following, here are the replies to individual remarks:

On line 103: The most recent work citing the model work mentioned by the reviewer is the comprehensive review on tribotesting of oscillating bearings of which the author takes pride in being a coauthor. However, the statement in this present work thakes a few shortcuts the deserve a more detailed explanation. Most notably, it has to be said that the risk of wear initiation is largely dependent on the lubricant and the wear onset to a final inoperability of the pitch bearing involves abrasive processes which might significantly influence the speed of the material removal and are not fully explored in models. The lines will be changed accordingly in the updated version of the document.

Line 181 (Q-calculation) The reviewer is perfectly right and the author a bit embarrassed that he forgot to mention this only looks at one of two axial rows. The paper will be updated accordingly.

Line 20: The author agrees and will update the final paper accordingly.

Line 29 to 54: The author agrees and will update the final paper accordingly.

Line 78: The author agrees and will update the final paper accordingly.

Line 80: The author agrees and will update the final paper accordingly.

Line 85: The author agrees and will update the final paper accordingly.

Line 131: Thank you very much for seeing this. Wrong latex command used. The author agrees and will update the final paper accordingly.